# The Vaccine Hesitancy Profiles and Determinants of Seasonal Influenza among Chinese Community Healthcare Workers: A Cross-Sectional Study

**DOI:** 10.3390/vaccines10091547

**Published:** 2022-09-16

**Authors:** Xianxian Yang, Wenge Tang, Qiang Tan, Deqiang Mao, Xianbin Ding

**Affiliations:** Department of Non-Communicable Disease Control and Prevention, Chongqing Center for Disease Control and Prevention, Chongqing 400042, China

**Keywords:** vaccine hesitancy, influenza, community healthcare workers, determinants, Chongqing, China

## Abstract

This paper is an evaluation of seasonal influenza vaccination hesitancy (IVH) and its determinants among community HCWs in Chongqing, a city in southwest China. Methods: A cross-sectional survey of 1030 community HCWs with direct or indirect patient contact was conducted from July to September 2021 using a self-administered electronic questionnaire. Possible factors for IVH among community HCWs were investigated by multivariable logistic regression to yield adjusted odds ratios (ORs) and 95% confidence intervals (CIs). Results: Overall, 46.2% of community HCWs were vaccinated in the 2020–2021 season, while 65.8% of community HCWs had IVH. “Don’t know the coverage in China” (OR: 1.46, 95% CI: 1.01–2.11; 40-year-old group OR: 3.02, 95% CI: 1.92–4.76), “complacency” (OR: 4.55, 95% CI: 3.14–6.60) were positively related with having IVH. The community HCWs that had a history of influenza vaccination (OR: 0.67 95% CI: 0.48–0.95) and groups with confidence and convenience (OR: 0.08, 95% CI: 0.06–0.12; OR: 0.34, 95% CI: 0.23–0.52, respectively) were more likely to completely accept vaccination. Conclusions: Measures such as improving the awareness and knowledge of influenza and vaccination and expanding the free vaccination policy, combined with improving the convenience of the vaccination service, will promote increased seasonal influenza vaccination-coverage in community HCWs in Chongqing.

## 1. Introduction

Globally, influenza causes a highly contagious infection with a significant morbidity and mortality burden, which causes 3 to 5 million cases of severe illness, and 290,000 to 650,000 respiratory deaths annually [1,2]. Healthcare workers (HCWs) are at a high risk of contracting influenza because of their occupational exposure to infected patients and virus-contaminated surfaces. A meta-analysis of 29 global studies has shown that HCWs who are not vaccinated have a 3.4 times higher risk for influenza infection than healthy adults [3]. In addition, HCWs are continuously exposed to the risk of being infected by influenza viruses during their work, which may, thus, further transmit influenza to vulnerable patients. Annual influenza vaccination is an important strategy to prevent influenza, especially in the context of the COVID-19 pandemic. A systematic review and meta-analysis mentioned that vaccine efficacy for the influenza vaccine is <50% for the outcome of hospitalization and death in older adults [4]. However, when well-matched between the seasonal influenza vaccine strains and the epidemic strains, circulating in the population, vaccine efficacy for HCWs reached 90% [5,6]. The World Health Organization (WHO) recommends HCWs as one of the target groups to receive seasonal influenza vaccination [7]. Health authorities in more than 90 countries recommend influenza vaccination for HCWs worldwide [8]. The technical guidelines for seasonal influenza vaccination in China [9], issued annually by the Chinese Center for Disease Control and Prevention, recommend that HCWs are the priority target group for influenza vaccination during the COVID-19 pandemic. Despite the disease severity of influenza and availability of safe vaccines, influenza vaccination coverage in HCWs is low, posing a challenge to public health worldwide [10,11]. Vaccine coverage among HCWs in the United States surpassed 75% in the 2017–2018 season and up to 95% of HCWs have workplace vaccination requirements by their employers [12]. However, in many European countries, such vaccination remains below 30% [13], In China, a systematic review reported that the highest vaccination coverage rate during five epidemic seasons since 2010 was no more than 15% among HCWs [14].

Firstly, vaccine hesitancy (VH) was identified as one of the 10 threats to global health by the WHO, in 2019. The use of vaccines has reduced the burden of infectious disease historically; however, VH is fueling the re-emergence of vaccine-preventable diseases and vaccination rates worldwide are plummeting [15]. Similarly, influenza vaccine hesitancy (IVH) reduces the influenza vaccination coverage [16]. 

However, most of the studies [17,18,19] on IVH were conducted in Western and developed countries, so the effect of IVH on vaccination coverage among community HCWs in China still remains unknown. In this study, we investigated the profiles of IVH and its related determinants in community HCWs in a southwest Chinese city, to explore the characteristics for a better understanding of IVH and to provide suggestions for future intervention.

## 2. Materials and Methods

### 2.1. Study Design and Data Collection

From July to September 2021, we conducted a cross-sectional design study to collect self-reported data through an online survey in six districts/counties from Chongqing (the districts and counties are of the same administrative level in Chongqing). A stratified cluster random sampling method was used to select six districts/counties based on geographic location and population composition. Then, we randomly selected 10 street (township) community health service centers (CHSCs) in each district/county. 

The inclusion criteria were (1) all community HCWs in 60 CHSCs and (2) age of 22 years or older. We defined community HCWs as those who worked at community health centers providing primary health services, including general practitioners, public health physicians, and nurses working on the front line. Community HCWs enrolled into this study with no incentives for participation and completed self-administered electronic questionnaires. 

The minimum required sample size of 1067 respondents was calculated using the formula: N = (Z^2^ × *P* × (1 – *P*))/d^2^, where Z = value from standard normal distribution corresponding to the desired confidence level (Z = 1.96 for 95% CI), *P* is the estimated proportion (*P* = 50%), and d is the desired precision of estimate (margin of error) (d = 3%). 

The electronic questionnaire included detailed questions on socio-demographic status (age, sex, residence, years engaged in medical service, education level, highest degree major, professional qualifications, and self-reported health condition), knowledge, attitudes, practice related to influenza and influenza vaccine, and influenza vaccine hesitancy. More details are provided in the Appendix A questionnaire.

### 2.2. Survey Instrument

The WHO Strategic Advisory Group of Experts (SAGE) developed the definition and determinants of the matrix of VH [20]. We defined IVH as: knowing that influenza vaccines and vaccination services are available, one is not completely confident whether to vaccinate or is still worried after vaccination. Options included: (1) completely reject, (2) reject but still considering, (3) have not decided yet or never thought about it, (4) accept but still considering, and (5) completely accept. Respondents who chose options 2, 3, or 4 were considered to have IVH.

The IVH scales for community HCWs included a 9-item section on the 3Cs model determinants of influenza vaccination (confidence, complacency, and convenience), assessed using a 5-point Likert scale (strongly disagree, disagree, neutral, agree, and strongly agree). The IVH scales passed the reliability and validity test in the study. Complacency dimension from “strongly agree” to “strongly disagree” with a score of 5–1 [21]. The other two dimensions are scored in reverse, from “strongly agree” to “strongly disagree” with a score of 1–5. The items that assessed confidence included: (1) the influenza vaccine is effective; (2) the influenza vaccine is safe; and (3) as for vaccination, I worry about flu vaccine for vaccine incidents. The items that assessed complacency included: (1) I have a high risk of getting influenza; (2) influenza is a big threat to my health; and (3) influenza vaccine is necessary to prevent me from getting influenza. The items that assessed convenience included: (1) the traffic from my house to the vaccination clinic is convenient; (2) I can afford the flu vaccine; and (3) I can easily find time to the clinic for influenza vaccination.

### 2.3. Study Measures

The major outcome measures in this study were: (1) necessary to get influenza vaccination against influenza, with responses dichotomized as strongly agree, indicating completely accepted vs. hesitancy (various degrees of hesitancy included neutral, indicating have not decided yet or never thought about it; agree to some extent, indicating accept but still considering; disagree to some extent, indicating reject but still considering) vs. strongly disagree, indicating completely rejected; (2) influenza vaccine uptake in the last influenza season (yes vs. no); (3) willingness to get influenza vaccination in the next influenza season (yes vs. no/maybe); and (4) willingness to recommend the vaccine to the patients (yes vs. no/maybe). The possible correlated factors (each 3Cs subscale) were dichotomized based on the mean value of each subscale variable as follows [22]: (1) confidence subscale: <10.5 vs. ≥10.5; (2) complacency subscale: <11.5 vs. ≥11.5; and (3) convenience subscale: <12.0 vs. ≥12.0. In the analysis [23], the median score of the 3Cs scale as a cutoff value and classified the subjects with IVH into two groups: mild hesitancy vs. severe hesitancy (<34.0 vs. ≥34). The covariates were sex, age (<30 years vs. 30–40 years vs. ≥40 years), residence (urban vs. rural), educational level (high/secondary school or lower vs. junior college vs. bachelor degree or above), professional category (clinical vs. traditional Chinese medicine vs. integrative medicine vs. nursing vs. preventive medicine/public health vs. other), years engaged in medical service (<10 years vs. ≥10 years), professional qualifications (primary or lower vs. middle title vs. senior title), and self-reported health condition (good vs. general vs. fair/poor). 

### 2.4. Statistics Analysis

Statistical analyses were performed using the Statistical Program for Social Sciences (SPSS) version 25.0 (IBM Corporation, New York, NY, USA). To explore the structure of IVH scales, Exploratory Factor Analysis (EFA) was conducted on the samples using Principal Axis Factoring with orthogonal rotation (varimax). Cronbach’s α was calculated, and factors with eigenvalues greater than one were extracted to determine the internal consistency. Descriptive statistical method was used to calculate vaccination coverage and the number of respondents based on reasons for and against vaccination. Categorical variables were compared using the Pearson chi-squared test. The independent sample t-test was used to evaluate the differences between vaccine outcome questions (IVH) to examine the criterion validity of the scale. Logistic regression analysis was used to explore relative factors of IVH and acceptance, after adjusting for potential confounding variables based on the odds ratios (OR) and 95% confidence interval (CI). Participants who completely accepted influenza vaccination were considered as the reference group, and vaccine hesitancy was taken as the dependent variable in the analysis. Potential confounding factors included demographic characteristics, knowledge, experience, and the score of each dimension of the “3Cs” model. Subgroup analyses were conducted through an independent sample t-test, according to the characteristics of IVH in each item between mild and severe hesitancy. All statistical tests were two-sided, and a *p* value < 0.05 was considered statistically significant.

### 2.5. Ethical Approval 

The study protocol and questionnaire were approved by the Research Ethics Committee of the Chongqing Centre for Disease Control and Prevention (2021(001)). The participants were reassured of the confidentiality of the collected information and signed informed consent. 

## 3. Results

### 3.1. Demographics

Overall, 1200 community HCWs were enrolled into this study. Of them, 1142 community HCWs completed the survey, so the response rate of this survey was 95.2%. However, 112 (9.8%) of 1142 eligible participants were excluded from the analysis because of failing to complete all items or pass the quality control questions.

In total, 1030 community HCWs were enrolled into this analysis, of which 54.1% of participants were in the age group between 20 and 39 years old. There were 293 males and 737 females, and the ratio of males to females was approximately 0.4:1. The proportion of years in medical service was 62.5% for 10 years and above, 45.3% of participants lived in an urban area, and 48.3% of participants had a college degree. There were significant differences in the characteristics of gender, age, educational level, and service year between the vaccine hesitancy subgroups (*p* < 0.05) (Table 1).

### 3.2. IVH

Only 25 (2.4%) community HCWs completely rejected the vaccination; 677 (65.8%) HCWs had IVH; and 328 (31.8%) HCWs completely accepted vaccination (Figure 1).

### 3.3. IVH Scores of all Variables of the “3Cs” Model

For the community HCWs’ IVH scale, the KMO measure of sampling adequacy was 0.894, and Bartlett’s test of sphericity (*p* < 0.001) indicated that sufficient correlations among the variables existed, allowing the study to proceed. EFA identified two factors with eigenvalues greater than one, explaining 59.50% of the common variance of nine items. All the standardized loadings were >0.6, and no cross-loading was >0.4, indicating that all items were significant. Finally, a reliability analysis revealed that the Cronbach’s α was 0.849. The t-test was performed on the scores of each dimension of the “3Cs” model, according to whether there was influenza vaccine hesitation or not (Table 2). There was a significant difference in each dimension of the “3Cs” model between the vaccine hesitancy subgroups (*p* < 0.05). For the community HCWs that were not IVH compared to those that were IVH, there was more trust, less complacency, and more perceived convenience and severity of influenza, as well as more trust in vaccine safety, effectiveness, and in vaccine delivery systems, so vaccination was considered more necessary and convenient.

### 3.4. Criterion Validity of the Scales

Community HCWs who completely accepted vaccination had a higher IVH score than those with hesitancy (Mean: 34.8 vs. 32.4, *p* < 0.001). HCWs who had received influenza vaccination before reported a higher IVH score than those who had no experience (Mean: 34.7 vs. 33.2, *p* < 0.001) (Table 3). 

### 3.5. Subgroups and Characteristics of IVH

We used the median score (34) of the 3Cs scale as a cutoff value and classified the subjects with IVH into two groups: those with mild hesitancy and severe hesitancy. Each item of the scales was different statistically between two groups (*p* < 0.001).

Of the nine items for HCWs IVH, the “impact of vaccine incidents”, “think HCWs have a high probability of getting flu”, “flu vaccine is necessary”, and “flu is a big threat to my health” were the leading and most important (Figure 2).

### 3.6. Determinants of IVH

Table 4 shows the result of multivariate logistic regression model on factors affecting IVH of community HCWs in Chongqing city during the 2020–2021 season. For community HCWs, variables of “don’t know the flu vaccine vaccination rates in China” (OR: 1.46, 95% CI: 1.01–2.11) and higher age (as compared with age group under <30 years old, age group of 40 years old and above (OR: 3.02, 95% CI: 1.92–4.76) and “complacency group” (OR: 4.55, 95% CI: 3.14–6.60) were positively related with having IVH. The HCWs who had influenza vaccination in the past year (OR: 0.67 95% CI: 0.48–0.95) and groups with high confidence in the efficacy and safety of vaccines and convenience (OR: 0.08, 95% CI: 0.06–0.12; OR: 0.34, 95% CI: 0.23–0.52 respectively) had a positive attitude toward vaccination, so were more likely to completely accept vaccination.

### 3.7. Influenza Vaccination Status, Willingness to Be Vaccinated, and Recommend Vaccination to Patients

Overall, 46.2% (95% CI: 43.2%–49.3%) of the community HCWs confirmed being vaccinated the influenza vaccine in the 2020–2021 season. In total, 635 (61.7%) respondents in the community HCWs were willing to be vaccinated against influenza during the 2021–2022 season, while 959 (93.1%) of the community HCWs were willing to recommend the influenza vaccine to their patients (Figure 3).

Out of 395 respondents who would not be vaccinated during the 2021–2022 season, the most common reason was that who believing vaccination was unnecessary, accounting for 60.8% (240/395) of respondents, followed by the vaccines not being free (35.9%, 142/395). The third reason was “the virus mutates quickly, concerns about inoculation effect”, which accounted for 25.1% (99/395) (Figure 4).

The reasons for not recommending the influenza vaccine to patients among the 71 community HCWs included: concerns about patients misunderstanding the selling of vaccines (63.4%), uncertainty of vaccine effectiveness for their patients (35.2%), HCWs consider that patients do not accept their recommendation on vaccines (16.9%), and do not know the contraindications of the influenza vaccination and dare not recommend it to patients (14.1%) (Figure 5). Furthermore, the proportion of those who recommended the seasonal influenza vaccine to patients was 64.8% in the vaccinated group, which was statistically higher than the 29.0% in the unvaccinated group during the 2021–2022 season (χ^2^ = 56.712, *p* < 0.0001).

## 4. Discussion

To the best of our knowledge, this is the first study to assess the profiles of IVH and its related determinants among community HCWs during the COVID-19 pandemic in mainland China. In 2009, China implemented a new round of healthcare reform to strengthen primary care networks, in which the development of Community Health Services (CHS) was the emphasis. Community HCWs provide primary healthcare, disease prevention, and health education, in the relevant jurisdiction to a key population, targeting pregnant women, young children, older adults, and adults with chronic diseases residing in the community [24]. Therefore, community HCWs can not only be one of the high-risk groups for influenza but also be trust messengers to promote influenza vaccine acceptance for fragile populations. Recommendations for seasonal influenza vaccination by HCWs is an effective way to improve influenza vaccine uptakes [25,26,27]. In addition, a study conducted by Song et al. indicated that community HCWs played a role in increasing influenza vaccine uptakes among high-risk groups in China [28]. Therefore, it has more practical guiding significance to explore the related factors for IVH among Chinese community HCWs. 

Hesitancy profile and determinants are related to the country and the environment globally. Similarly to previous studies [23,29,30], we describe IVH as an intention/attitude between complete rejection and acceptance, which was assessed by one item based on the “3Cs” model. It is considered one of the most useful models for analyzing vaccine hesitancy [31]. The findings of this survey indicated that 65.7% of the community HCWs, who will be at risk of influenza virus infection, reported IVH, indicating that challenges arise among those who fail to seek the vaccination service despite the full acceptance in Chongqing. Their main concerns with the influenza vaccine were safety, efficacy, and side effects. A cross-sectional study [32] in Oman was consistent with our findings. Moreover, studies [32,33] from the general public on IVH have found that recommendations by healthcare providers are associated with lower odds for IVH. However, if community HCWs continue to remain hesitant towards influenza vaccines, it is unlikely that they would recommend these vaccines to the general public and ensure mass vaccinations with the available influenza vaccines.

As vaccine safety has been one of the most important predictors of hesitancy worldwide, studies [21,34] have shown that when people perceived greater trust in the safety and efficacy of the vaccine and the system for delivering it, the less hesitant to be vaccinated. When people perceived a higher necessity for vaccination and a severity of influenza infection, they are less complacent and less hesitant to be vaccinated. Moreover, when people perceived a higher convenience for vaccination, the less hesitant they are to be vaccinated, similar to the current study. The results of multivariate logistic regression analysis in this study showed that complacency was a risk factor for IVH, while trust and convenience were protective factors for IVH. In addition, this study showed that the community HCWs who had been vaccinated with the influenza vaccine in the past year may reduce the hesitancy as well as reduce the infection risk. The main reasons include their own previous vaccination protection experience, surrounding people or family, and reduction in illness. A previous study [35] showed that influenza vaccination history was an important factor influencing influenza vaccination among medical staff, people who have been vaccinated with the seasonal influenza vaccine are more concerned about influenza illness, and people place a greater emphasis on using vaccines to protect themselves. However, the participants did not know the vaccination rates in China relate to the independent factors affecting the vaccination status of community HCWs, according to our survey. Thus, the poor knowledge of influenza vaccines in community HCWs was a huge challenge, because HCWs with good levels of knowledge are more likely to recommend the influenza vaccine to their patients [36]. The present study suggested that to increase the coverage rates of influenza vaccination, increasing community HCWs’ confidence in influenza vaccination, providing more convenient vaccination services, and reducing vaccine-related complacency are all necessary. Furthermore, educational campaigns should be launched to improve community HCWs’ awareness and knowledge of influenza vaccination. It is reported [32,36] that knowledge was a critical factor that influenced HCWs’ willingness to receive influenza vaccination as well as recommend it to patients. In the long run, increasing influenza vaccine acceptance and uptakes among community HCWs should be a key component of pandemic preparedness, both to protect them and to promote vaccination among the public during a pandemic [37]. 

A systematic literature review [38] showed that HCWs with >10 years of service were significantly more likely to recommend influenza vaccination to their patients; however, no association was found between years of service and vaccine uptake or IVH in the study. Similar findings were observed in a cross-sectional study [32] in Oman, where the length of service did not correlate with greater compliance with vaccination. Studies about age as the related influencing factor of VH have been inconsistent [22,23]. However, multinomial regression analysis showed that advanced-age people have higher IVH than younger ones in the study, possibly because they perceived influenza to be a minor illness and thought antiviral drugs for influenza can effectively prevent and treat the illness.

Coverage rates of the seasonal influenza vaccination in HCWs vary widely in different countries and regions. During the 2014–2015 influenza season, it was reported that influenza vaccine ranging from 2.6% to 99.5% among HCWs in 26 (56%) European countries [39]. However, a lot of survey data showed that the coverage rates among HCWs in China were generally low, despite the availability of the vaccines to all HCWs at their workplaces. The coverage among HCWs in two hospitals that have a free vaccination policy was 30.5% and 25.9%, respectively, in the 2018–2019 seasons in Xining city [40]. The coverage of the 2020–2021 season among community HCWs in Chongqing was 46.2%, according to our surveys. Vaccination services can be delivered by general practitioners in many countries; however, in China, they are delivered by dedicated vaccinators at vaccination clinics held in community health centers [41]. So, the community HCWs not only could obtain accurate vaccination-related information in the first time but also could complete vaccinations in their workplace. Secondly, the governments of some districts and counties of Chongqing introduced a free influenza vaccinations policy for government cadres and staff, including medical staff. Thirdly, the community HCWs have vaccination distribution responsibilities, so they play an active role in vaccination-related health education or consultations. Finally, the coverage rate of influenza vaccination in community HCWs of the 2020–2021 season was surveyed during the COVID-19 pandemic in mainland China. At that time, COVID-19 vaccines were not widely available, so most people chose the seasonal influenza vaccination against COVID-19. Furthermore, a recent study [42] showed that expanding influenza vaccination coverage has a role in the management of respiratory outbreaks such as COVID-19. From this point of view, increasing influenza coverage rate and incorporating recommendation behaviors among community HCWs could become helpful in the fight against respiratory outbreaks.

The findings of the survey showed that the most common reasons for not being vaccinated during the 2021–2022 were necessity and the price of vaccination. Previous studies [40,43] have also found that cost is a common barrier to receive an influenza vaccination, especially in places where vaccination is not covered by health insurance. However, a limited number of regions in Chongqing have implemented a free influenza vaccination policy. This indicated that in order to achieve a relatively high level of coverage in community HCWs, in the absence of free policy legislative or mandatory vaccination requirements, it is very important to provide free vaccinations and improve the awareness of influenza and the influenza vaccine in community HCWs [44,45,46]. We can improve the knowledge level of community HCWs to enhance their understanding of the efficacy, benefit, and safety of vaccines, and eliminate their distrust of and hesitation about vaccines, so as to improve their vaccination compliance [47,48,49]. 

A large number of studies [28,50,51,52] have shown that HCWs’ recommendations were associated with an increased coverage rate, especially among high-risk groups. However, it is not common for medical staff to recommend influenza vaccination to high-risk groups in China. It is reported that only 8% of HCWs recommended influenza vaccination to their patients in Qingdao, and only 4% of pregnant women and 27% guardians of young children received a HCW’s recommendation for influenza vaccination [28]. Several determinants may prevent HCWs in China from routinely recommending influenza vaccination in China. In our study, the major reason was that community HCWs were concerned about patients misunderstanding the selling of vaccines. Moreover, community HCWs may be unwilling to recommend optional vaccines not included in the National Immunization Program (NIP), such as the influenza vaccine, to decrease their risk of being held liable for adverse events. Moreover, community HCWs were concerned about vaccine safety and effectiveness and perceived vaccination as unnecessary to their patients. Further, we found that there were differences in the perception of the influenza vaccination and in recommending it between the two groups. Community HCWs in the vaccinated group have a higher potential to recommend the influenza vaccination to their patients, as indicated in previous studies [14,40,53]. Therefore, we recommend providing timely and accurate information about seasonal influenza vaccination for community HCWs, to make sure that they have the knowledge required to make effective recommendations. The increased awareness and vaccination of community HCWs could further encourage clinicians to recommend influenza vaccination, to achieve higher influenza vaccination rates in high-risk groups in China. 

However, our study had several limitations. First, the cross-sectional research limited our exploration of the causal relationship between IVH and determinants, and the small sample size in the analysis may lead to the weak power of IVH and, thus, the generality may be hindered. Second, the results were self-reported, and vaccination records were not verified further. However, we think community HCWs are mostly professional, so the possibility of false reports was relatively low. Third, the participants in the current study were from one province in the southwest region of China, and, thus, the conclusions for IVH may not be generalized to other areas in the country. Fourth, potential selection bias may also arise. Since the two community health service centers that implemented the free policy also provided centralized vaccination services, the impact of the free policy and vaccination service could not be evaluated separately.

## 5. Conclusions

This is the first study to assess the knowledge, attitudes, and practices of community HCWs regarding the seasonal influenza vaccine in southwest of China. The major findings of this study can be summarized by the following three points. First, influenza vaccine uptake was 46.2% during the last influenza season (2020/2021), which is slightly higher than the rates observed in other cities in China [40]. Second, the accessibility of vaccination services leads to an evident improvement in influenza vaccination coverage in the community HCWs surveyed in our study. However, 65.8% of the community HCWs were hesitant about influenza vaccination in our study. Third, the determinants of seasonal influenza vaccine hesitancy include age, knowledge of influenza and influenza vaccine, and influenza vaccination history, while confidence, complacency, and convenience were significantly correlated with influenza vaccine acceptance among Chinese HCWs.

In order to increase seasonal influenza vaccination coverage in community HCWs, it is crucial to improve the community HCWs’ personal confidence, their awareness of influenza and vaccination, and their accurate knowledge of influenza vaccines, to engage them in activities targeting vaccine hesitancy among their patients. In addition, it is crucial to expand the free vaccination policy and improve the accessibility of vaccination service. Furthermore, we should continue to explore effective interventions to increase the coverage rate of community HCWs, in order to promote the vaccination of the general population at the individual and social levels.

## Figures and Tables

**Figure 1 vaccines-10-01547-f001:**
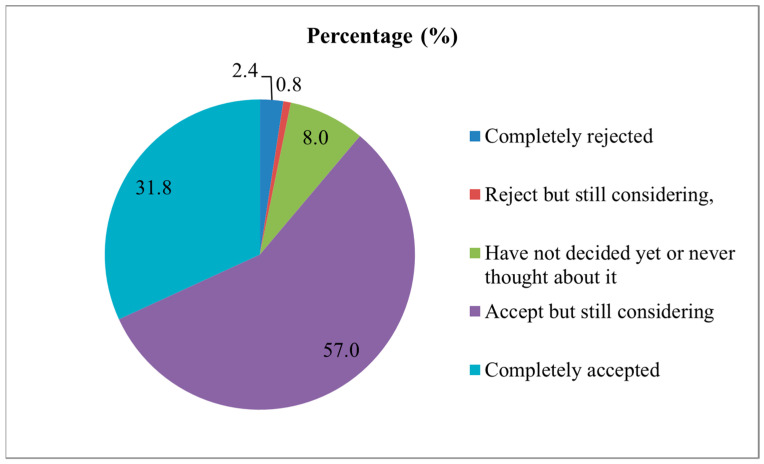
Proportion of influenza vaccine hesitancy in community healthcare workers.

**Figure 2 vaccines-10-01547-f002:**
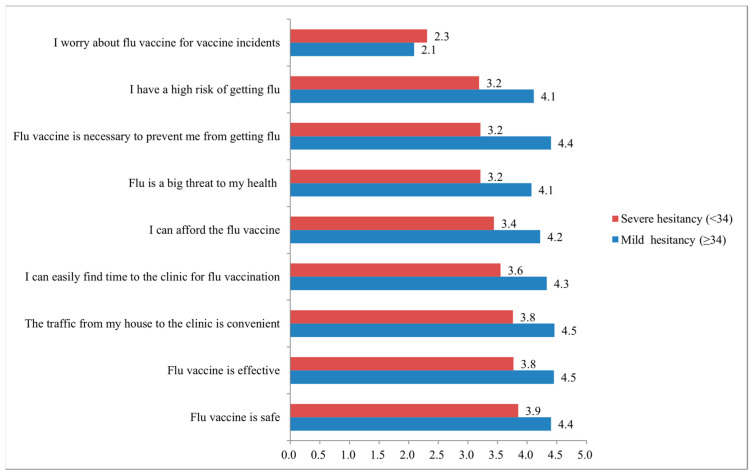
Subgroups and characteristics of influenza vaccine hesitancy (IVH): difference in each item between mild and severe hesitancy was significant (*p* < 0.001) by independent sample *t*-test.

**Figure 3 vaccines-10-01547-f003:**
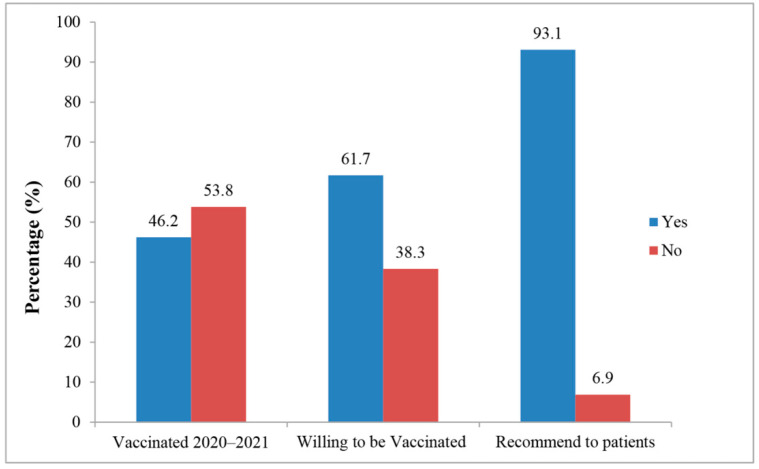
The proportion of community HCWs vaccinated, willing to vaccinate, and willing to recommend the influenza vaccine to patients.

**Figure 4 vaccines-10-01547-f004:**
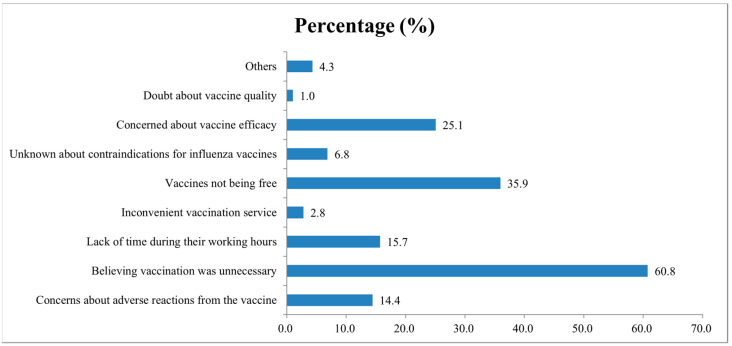
Proportion of reasons for not receiving the influenza vaccination among healthcare workers during the 2021–2022 season.

**Figure 5 vaccines-10-01547-f005:**
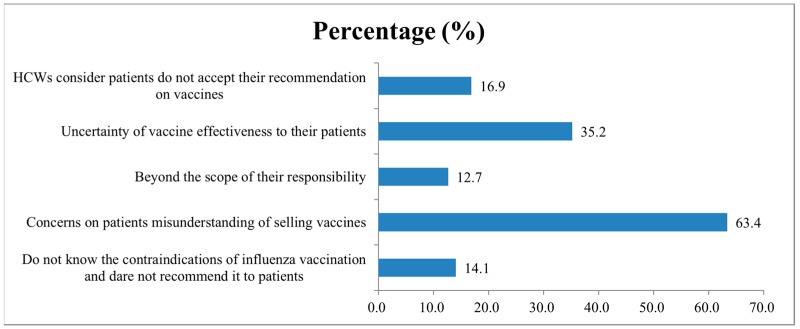
Reasons for not recommending the influenza vaccination to patients among community HCWs.

**Table 1 vaccines-10-01547-t001:** Differences in influenza vaccine hesitancy between different demographic characteristics.

Variables	Sample Size, *n* (%)	Positive		2	*p*
		N	%		
Gender				6.423	0.011
Male	293 (28.5)	210	31.0		
Female	737 (71.5)	467	69.0		
Age (years)				21.378	<0.001
<30	240 (23.3)	137	20.2		
30–	317 (30.8)	195	28.8		
40–	473 (45.9)	345	52.0		
Residence				0.019	0.89
Urban	467 (45.3)	308	45.5		
Rural	563 (54.7)	367	54.5		
Education level				15.036	<0.001
High/secondary school or lower	155 (15.1)	121	17.9		
Junior college	377 (36.6)	251	37.1		
Bachelor degree or above	698 (48.3)	305	45.0		
Professional category				15.229	0.009
Clinical	323 (31.4)	217	32.1		
Traditional Chinese medicine	85 (8.2)	52	7.7		
Integrative medicine	116 (11.3)	93	13.7		
Nursing	341 (33.1)	216	31.9		
Preventive medicine/public health	45 (4.4)	28	4.1		
Other	120 (11.6)	71	10.5		
Years engaged in medical service					
<10	386 (37.5)	221	32.6	19.681	<0.001
≥10	644 (62.5)	456	67.4		
Professional qualifications				1.506	0.68
Primary or lower	693 (67.3)	451	66.6		
Middle title	265 (25.7)	181	26.7		
Senior title	72 (7.0)	45	6.7		
Self-reported health condition				5.608	0.061
Good	763 (74.0)	488	72.1		
General	187 (18.2)	128	18.9		
Fair/Poor	80 (7.8)	61	9.0		

**Table 2 vaccines-10-01547-t002:** Average scores of 1030 respondents in each dimension of the “3Cs” model.

Dimension	Item	Total (*n* = 1030)	Vaccine Hesitancy		*t*	*p*
			Yes (*n* = 677)	No (*n* = 353)		
Confidence		10.5 ± 1.1	10.2 ± 0.9	11.1 ± 1.3	11.378	<0.001
Effective	Flu vaccine is effective.	4.1 ± 0.7	3.9 ± 0.5	4.6 ± 0.9	13.427	<0.001
Safety	Flu vaccine is safe.	4.1 ± 0.6	3.9 ± 0.5	4.6 ± 0.6	16.657	<0.001
Trust in the vaccine delivery system	I worry about flu vaccine for vaccine incidents.	2.2 ± 0.8	2.3 ± 0.6	1.9 ± 0.9	−7.997	<0.001
Complacency		11.5 ± 1.8	10.8 ± 1.4	12.7 ± 1.9	16.391	<0.001
Importance	I have a high risk of getting flu.	3.7 ± 0.8	3.4 ± 0.7	4.1 ± 0.9	12.125	<0.001
Severity	Flu is a big threat to my health.	3.7 ± 0.8	3.5 ± 0.7	4.1 ± 0.9	12.012	<0.001
Necessity	Flu vaccine is necessary to prevent me from getting flu.	4.1 ± 0.6	3.9 ± 0.5	4.5 ± 0.7	15.257	<0.001
Convenience		12.0 ± 1.8	11.3 ± 1.5	13.2 ± 1.8	16.856	<0.001
	The traffic from my house to the clinic is convenient.	4.1 ± 0.7	3.9 ± 0.6	4.6 ± 0.6	15.829	<0.001
	I can afford the flu vaccine.	3.9 ± 0.8	3.6 ± 0.7	4.2 ± 0.8	11.971	<0.001
	I can easily find time to the clinic for flu vaccination.	4.0 ± 0.7	3.8 ± 0.6	4.4 ± 0.7	13.827	<0.001

**Table 3 vaccines-10-01547-t003:** Influenza vaccine hesitancy (IVH) scale scores between different groups of subjects.

Variables	*n*	Mean	SD	*t*	*p*
Future influenza vaccination intention (N = 831)					
Hesitant	196	32.4	3.3	8.264	<0.001
Completely agree	635	34.8	3.7		
Influenza vaccination experience (N = 831)					
Never vaccinated	554	33.2	3.6	6.462	<0.001
Vaccinated before	476	34.7	3.9		

**Table 4 vaccines-10-01547-t004:** Determinants of influenza vaccine hesitancy in community healthcare workers.

Variables	Unadjusted OR (95% CI)	*p*	Adjusted OR (95% CI)	*p*
Gender				
Male	1.00			
Female	0.69 (0.51–0.94)	0.018		
Age (years)				
<30	1.00		1.00	
30–	1.20 (0.84–1.70)	0.314	1.51 (0.95–2.41)	0.084
40–	1.99 (1.43–2.78)	<0.001	3.02 (1.92–4.76)	<0.001
Residence				
Urban	1.00			
Rural	1.05 (0.80–1.37)	0.731		
Educational level				
High/secondary school or lower	1.00			
Junior college	0.58 (0.37–0.90)	0.016		
Bachelor degree or above	0.47 (0.31–0.72)	<0.001		
Years engaged in medical service				
<10	1.00			
≥10	1.76 (1.34–2.31)	<0.001		
Professional qualifications				
Primary or lower	1.00			
Middle title	1.17 (0.86–1.60)	0.315		
Senior title	0.86 (0.52–1.44)	0.572		
Self-reported health condition				
Good	1.00			
General	1.24 (0.87–1.76)	0.234		
Fair/Poor	1.68 (0.98–2.87)	0.059		
Knowledge				
The flu vaccine vaccination rates in China (ref = know)				
Don’t know	1.36 (1.02–1.81)	0.036	1.46 (1.01–2.11)	0.044
Flu is different from common cold (ref = know)				
Don’t know	2.31 (1.34–3.98)	0.016		
The whole population is susceptible to flu (ref = know)				
Don’t know	2.73 (1.20–6.18)	0.016		
Flu can be spread through respiratory droplets, or through direct or indirect contact with mucous membranes such as the mouth, nose and eyes (ref = know)				
Do not know	1.17 (0.51–2.67)	0.72		
Experience				
Had a flu before (ref = No or not clear)				
Yes	1.00 (0.65–1.54)	0.99		
Had flu-like symptoms before (ref = No or not clear)				
Yes	0.87 (0.63–1.20)	0.395		
Had been vaccinated influenza vaccine in the past year (ref = No)				
Yes	0.47 (0.36–0.61)	<0.001	0.67 (0.48–0.95)	0.024
Communities (CHSC) have promoted flu vaccination (ref = No or not clear)				
Yes	0.64 (0.43–0.96)	0.029		
“3Cs”				
Complacency (ref = No)				
Yes	5.39 (3.96–7.34)	<0.001	4.55 (3.14–6.60)	<0.001
Confidence (ref = No)				
Yes	0.09 (0.06–0.12)	<0.001	0.08 (0.06–0.12)	<0.001
Convenience (ref = No)				
Yes	0.22 (0.16–0.32)	<0.001	0.34 (0.23–0.52)	<0.001

Note: (i) The score of each dimension of the “3Cs” model was divided into two groups according to the average value, and vaccine hesitancy was taken as the dependent variable in the analysis; (ii) dependent variables used in multivariate logistic regression (0 = completely agree, 1 = IVH). CI: confidence interval; CHSC: community health service center.

## Data Availability

The data presented in this study are available on request from the corresponding author (Xianbin Ding).

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
