# Peer review of "The Vaccine Hesitancy Profiles and Determinants of Seasonal Influenza among Chinese Community Healthcare Workers: A Cross-Sectional Study"

_vaccines, 2022, doi:10.3390/vaccines10091547_

Round 1

Reviewer 1 Report

Introduction:

Please mention and cite the literature that the vaccine efficacy for the influenza vaccine is < 50% for the outcome of hospitalization and death.

Example citation:

Domnich A, de Waure C. Comparative effectiveness of adjuvanted versus high-dose seasonal influenza vaccines for older adults: A systematic review and meta-analysis. Int J Infect Dis. 2022 Jul 22:S1201-9712(22)00443-X. doi: 10.1016/j.ijid.2022.07.048. Epub ahead of print. PMID: 35878803

Introduction:  please mention that vaccine hesitancy is justified when vaccines have unacceptable vaccine efficacy < 50%.

Discussion

Please indicate that the minimal effectiveness of the influenza vaccine suggests it should be restricted to the very high risk elderly and that mass indiscriminate vaccination of health care workers is not justified.  Furthermore, public programs to vaccination younger individuals and children should be discontinued based on very poor vaccine efficacy.

Author Response

Seasonal Influenza Vaccine Hesitancy Profiles and Determinants among Chinese Community Healthcare Workers: A Cross-sectional Study

(Manuscript ID: vaccines-1891892)

Dear reviewer,

Thank you for your valuable comments. We have modified the manuscript, and revisions were highlighted in red to make sure changes could be easily visible to you. Please see our responses to the comments.

Response to Reviewer 1 Comments

Point 1: Introduction: Please mention and cite the literature that the vaccine efficacy for the influenza vaccine is < 50% for the outcome of hospitalization and death.

Example citation: Domnich A, de Waure C. Comparative effectiveness of adjuvanted versus high-dose seasonal influenza vaccines for older adults: A systematic review and meta-analysis. Int J Infect Dis. 2022 Jul 22:S1201-9712(22)00443-X. doi: 10.1016/j.ijid.2022.07.048. Epub ahead of print. PMID: 35878803

Response 1: Thanks for sharing the updated reference. Efficacy is the degree to which a vaccine prevents disease, and possibly also the transmission, under ideal and controlled circumstances comparing a vaccinated group with a placebo group. Effectiveness meanwhile refers to how well it performs in the real world. In general, there is considerable variability in the efficacy and effectiveness of influenza vaccines in different seasons, different population groups, and different vaccines. A systematic review comparing cell-based TIVs with placebo in adults aged 18–49 years found that overall vaccine efficacy against laboratory-confirmed influenza was 70% against any influenza subtype (95% CI: 61–77%), 72% against influenza A(H3N2) (95% CI: 39–87%) and 52% against influenza B (95%: CI 30–68%)[1] . HD-IIV3 was also more effective at preventing hospital admissions from all-causes (rVE = 8.4%, 95% CI: 5.7–11.0%), as well as influenza (rVE = 11.7%, 95% CI: 7.0–16.1%), pneumonia (rVE = 27.3%, 95% CI: 15.3–37.6%), combined pneumonia/influenza (rVE = 13.4%, 95% CI: 7.3–19.2%) and cardiorespiratory events (rVE = 17.9%, 95% CI: 15.0–20.8%)[2]. We have mentioned and cited the literature that the vaccine efficacy for the influenza vaccine is < 50% for the outcome of hospitalization and death according to the suggestion. Please see the “Introduction” section of the revised manuscript. 

Point 2: Introduction: Please mention that vaccine hesitancy is justified when vaccines have unacceptable vaccine efficacy of < 50%.

Response 2: Vaccination is considered the primary preventive measure to prevent influenza. Vaccine hesitancy is complex and context-specific, varying across time, place, and vaccines. It’s not only affected by vaccine efficacy or vaccine effectiveness, but also influenced by a range of factors, such as safety, knowledge, policy, ect. Furthermore, we have searched for a lot of literature, but previous studies did not indicate that vaccine hesitancy is justified when vaccines have unacceptable vaccine efficacy of < 50%. We would appreciate it if you could share the reference for us. Maybe this point can be further explored in our future studies.

Point 3: Discussion: Please indicate that the minimal effectiveness of the influenza vaccine suggests it should be restricted to the very high-risk elderly and that mass indiscriminate vaccination of health care workers is not justified. Furthermore, public programs to vaccination younger individuals and children should be discontinued based on very poor vaccine efficacy.

Response 3: Risk groups for influenza include groups at particular risk of developing severe disease resulting in hospitalization or death and those at increased risk of exposure to or transmission of influenza virus. The latter group includes health workers. Based on the latest World Health Organization (WHO) recommendations[3], the following groups should be prioritized for annual influenza vaccination before the beginning of the influenza season: (1) HCWs, (2) pregnant women, (3) individuals with certain chronic diseases for more than six months, (4) elderly people over the age of 65 years, (5) children aged six months to five years and (6) residents of institutions for older persons and the disabled. Health workers are at increased risk of contracting influenza and may further transmit influenza to vulnerable population groups. Vaccination of children can reduce the number of influenza or influenza-like illness and may reduce community transmission, including to vulnerable groups.

  In general, there is considerable variability in the efficacy and effectiveness of influenza vaccines in different seasons and different population groups. A systematic review comparing cell-based TIVs with placebo in adults aged 18–49 years found that overall vaccine efficacy against laboratory-confirmed influenza was 70% against any influenza subtype (95% CI: 61–77%), 72% against influenza A (H3N2) (95% CI: 39–87%) and 52% against influenza B (95% CI: 30–68%)[1].

Also, previous reviews[4] have found that the influenza vaccine is either cost-saving or has an acceptable cost-effectiveness ratio, at 75% vaccination coverage by discharge, vaccination was cost-saving from the healthcare payer perspective in adults ≥ 65 years and the ICER was $12,680/QALY (95% CI: 6,273-20,264) in adults 18-64 years and $2,400 (95% CI: -1,992-7,398) in all adults 18 + years. 

References

  1. Jordan, K.; Murchu, E.O.; Comber, L.; Hawkshaw, S.; Marshall, L.; O'Neill, M.; Teljeur, C.; Harrington, P.; Carnahan, A.; Pérez-Martín, J.J.; et al. Systematic review of the efficacy, effectiveness and safety of cell-based seasonal influenza vaccines for the prevention of laboratory-confirmed influenza in individuals ≥18 years of age. Rev Med Virol 2022, e2332, doi:10.1002/rmv.2332.
  2. Lee, J.K.H.; Lam, G.K.L.; Shin, T.; Samson, S.I.; Greenberg, D.P.; Chit, A. Efficacy and effectiveness of high-dose influenza vaccine in older adults by circulating strain and antigenic match: An updated systematic review and meta-analysis. Vaccine 2021, 39 Suppl 1, A24-a35, doi:10.1016/j.vaccine.2020.09.004.
  3. Vaccines against influenza: WHO position paper – May 2022
  4. Peasah, S.K.; Meltzer, M.I.; Vu, M.; Moulia, D.L.; Bridges, C.B. Cost-effectiveness of increased influenza vaccination uptake against readmissions of major adverse cardiac events in the US. PLoS One 2019, 14, e0213499, doi:10.1371/journal.pone.0213499.

Reviewer 2 Report

Thank you for the invitation to review this paper. The findings of this paper is of good interest. I have few comments;

1. The questionnaire used in this study was modified and translated, I am not sure that the authors have performed the reliability and validity of that tool. Please provide the english version of the questionnaire along with the manuscript file. Please provide the detailed methods of validation and translation in the method section of the manuscript. 

2. The method section of the manuscript is very simple. How the scoring was done and how the participants were classified into required outcomes are not clear in the methods.

3. The current study is the first one in the region, this statement should be provided at the start of the discussion section.

4. The conclusion section is very general and vague, please provide the major findings of this study in the conclusion section along with future directions for research and policies.

5. The manuscript should be improved for scientific writing.

Author Response

Seasonal Influenza Vaccine Hesitancy Profiles and Determinants among Chinese Community Healthcare Workers: A Cross-sectional Study

(Manuscript ID: vaccines-1891892)

Dear reviewer,

Thank you for your valuable comments. We have modified the manuscript, and revisions were highlighted in blue to make sure changes could be easily visible to you. Please see our responses to the comments.

Response to Reviewer 2 Comments

Point 1: The questionnaire used in this study was modified and translated, I am not sure that the authors have performed the reliability and validity of that tool. Please provide the English version of the questionnaire along with the manuscript file. Please provide the detailed methods of validation and translation in the method section of the manuscript. 

Response 1: We designed one scale to recognize the characteristics of IVH among the community HCWs, and the tool passed the reliability and validity test in the study. For the community HCWs’ IVH scale, the KMO measure of sampling adequacy was 0.894, and Bartlett’s test of sphericity (p < 0.001) indicated that sufficient correlations among the variables existed, allowing to proceed. EFA identified two factors with eigenvalues greater than one, explaining 59.50% of the common variance of 9 items. All the standardized loadings were >0.6 and no cross-loading was >0.4, indicating that all items were significant. Finally, a reliability analysis revealed that the Cronbach’s α was 0.849. Additionally, we have provided the English version of the questionnaire along with the manuscript file, and we have added the detailed methods of validation and translation in the “2.4 Statistics analysis” and “3.3. IVH scores of all variables in the “3Cs" model” sections of the revised manuscript. Thanks for your good suggestions.

Point 2: The method section of the manuscript is very simple. How the scoring was done and how the participants were classified into required outcomes are not clear in the methods.

Response 2: We have modified the “Method ” section to make it clearer. We have added how the scoring was done and how the participants were classified into required outcomes. Please see the paragraph in the “Survey Instrument” and “Study Measures” sections. Thanks.

Point 3: The current study is the first one in the region, this statement should be provided at the start of the discussion section.

Response 3: We have provided the statement at the start of the discussion section of the revised manuscript according to the suggestion. Thanks.

Point 4: The conclusion section is very general and vague, please provide the major findings of this study in the conclusion section along with future directions for research and policies.

Response 4: We have provided the major findings of this study in the conclusion section along with future directions for research and policies according to the suggestion. Please see the “conclusion” section of the revised manuscript.

Point 5: The manuscript should be improved for scientific writing.

Response 5: The revised manuscript have be improved for scientific writing according to each of the comments. 

Reviewer 3 Report

Thank you for the invitation. The authors have conducted a good analysis of VH among the Chinese population. I have a few suggestions that should be considered before making any decision. There is a need to elaborate on the components of the data collection form. The authors have used the 3C mode of VH, but it is not described in the method section. There is also a need to define the terms used in the analysis, for example; severe hesitancy, mild hesitancy, complete acceptance, complete rejection, and various degrees of hesitancy used in figure 1.

The quality of the figures should be improved. The authors are requested to provide reasoning for the risk factors of VH in their analysis. For example, why did advanced-age people have higher VH than younger ones? 

There are several grammatical and syntax errors in the manuscript. For example: But we think community HCWs are mostly professional, and the 366 possibility of the false reports was. I think this sentence is incomplete. The use of punctuation should be considered carefully. 

There is missing information on the translation and validation of the data collection form. Please provide an English version of the form as a supplementary file of the manuscript.

Author Response

Seasonal Influenza Vaccine Hesitancy Profiles and Determinants among Chinese Community Healthcare Workers: A Cross-sectional Study

(Manuscript ID: vaccines-1891892)

Dear reviewer,

Thank you for your valuable comments. We have modified the manuscript, and revisions were highlighted in red to make sure changes could be easily visible to you. Please see our responses to the comments.

Response to Reviewer 3 Comments

Point 1: There is a need to elaborate on the components of the data collection form. The authors have used the 3C mode of VH, but it is not described in the method section. There is also a need to define the terms used in the analysis, for example; severe hesitancy, mild hesitancy, complete acceptance, complete rejection, and various degrees of hesitancy used in figure 1.

Response 1: We have added elaborate on the components of the data collection form in the “2.1. Study design and data collection” section of the revised manuscript. Also, we have provided the English version of the questionnaire along with the manuscript file. Moreover, we have added a detailed description of the 3Cs mode of VH in the “2.2. Survey Instrument” section of the revised manuscript.We have defined the terms used in the analysis, please see “2.3. Study Measures” and “3.5. Subgroups and characteristics of IVH” sections of the revised manuscript.

Point 2: The quality of the figures should be improved. The authors are requested to provide reasoning for the risk factors of VH in their analysis. For example, why did advanced-age people have higher VH than younger ones? 

Response 2: We have improved the quality of the figures. In our study, it showed that advanced-age people have higher VH than younger ones, possibly because they perceived influenza to be a minor illness and thought antiviral drugs for influenza can effectively prevent and treat the illness. Furthermore, we have provided reasoning for the risk factors of VH in our analysis, please see the “Discussion” section of the revised manuscript.

Point 3: There are several grammatical and syntax errors in the manuscript. For example: But we think community HCWs are mostly professional, and the 366 possibility of the false reports was. I think this sentence is incomplete. The use of punctuation should be considered carefully. 

Response 3: Yes, this sentence is incomplete. We have made a slight change here “But we think community HCWs are mostly professional, and the possibility of the false reports was relatively low”. We have corrected some grammatical and syntax errors in the manuscript. Furthermore, our revised manuscript has been carefully proofread by an English-speaking person. Thanks.

Point 4: There is missing information on the translation and validation of the data collection form. Please provide an English version of the form as a supplementary file of the manuscript.

Response 4: We have provided an English version of the form as a supplementary file of the manuscript according to the suggestion. Thanks.
